# Multi-Modal and Multi-Task Transformer for Small Molecule Drug Discovery

Sai Krishna Sirumalla [* 1]   David S. Farina Jr [* 1]   Zhuoran Qiao [1]   Daniele A. Di Cesare [1]   Felipe Costas Farias [1]
Michael B. O'Connor [1]   Peter J. Bygrave [1]   Feizhi Ding [1]   Thomas Dresselhaus [1]   Marcelo G. P. de Lacerda [1]
Jason M. Swails [1]   Daniel Miles [1]   Matthew Welborn [1]   Frederick R. Manby [1]   Thomas F. Miller III [1]

## Abstract

We introduce a 1B-parameter transformer model pre-trained from scratch on 2.25 T tokens from a massive mixture of datasets centered around drug discovery. These datasets are heterogeneous, coming from dozens of sources and covering 15 data modalities. We demonstrate the model's capability on various molecular assay prediction tasks, including public benchmarks and internally generated holdouts from real-world drug discovery programs. Following parameter-efficient fine-tuning, the multi-modal transformer excels at multi-task predictions compared to strong molecular property prediction baselines including XGBoost and Chemprop.

## 1. Introduction

The process of discovering and developing new drugs is long and complex, spanning many years and involving many stages from initial target identification though clinical trials. Each stage of the pipeline features a wide array of experimental endpoints and generates diverse types of data, ranging from chemical structures and physicochemical properties to biological assay results and clinical outcomes. The sheer complexity and heterogeneity of these data make it challenging for practitioners to accurately design compounds and make informed decisions about which compounds to prioritize for further development.

Machine learning has long promised to change the way we discover drugs, and recently deep learning has seen enormous interest (Askr et al., 2023). However, its success has been limited by several key challenges related to the nature and quality of available data. The first major obstacle is the high cost of acquiring high-quality experimental data. While there are some large public databases available, such

as PubChem and ChEMBL, these resources often suffer from low data quality and from heterogeneity (Landrum & Riniker, 2024). The second major obstacle is the enormous number of data modalities in the biomedical field, ranging from assay measurements to x-ray crystal structures to clinicians' notes. Previous approaches have addressed these issues through careful data curation (Seidl et al., 2023) and modality-specific architectures (Liu et al., 2023a; Edwards et al., 2022; Liu et al., 2023b). However, the operational complexity of these approaches has limited their respective scopes to a small number of data sources and/or data modalities.

In this work, we propose to train a single model on dozens of public and private data sources, and 15 data modalities. We eschew careful data curation and bespoke model architecture, choosing instead to represent all data as strings of tokens, and to train a large transformer model. We report results for a 1B-parameter multi-modal transformer (Vaswani et al., 2017; Touvron et al., 2023a;b; Radford et al., 2018; 2019; Brown et al., 2020; Achiam et al., 2023; Chowdhery et al., 2023; Black et al., 2022; Yasunaga et al., 2022) model trained on 2.25T tokens.

## 2. Data Pipeline

Biomedical data is extremely heterogeneous. Pre-training a large transformer model requires an enormous corpus of data. To effectively train a large multi-modal transformer model for drug discovery, we developed a data pipeline to standardize, clean, and transform a large corpus of biomedical data from many sources and of various types into streams of tokens suitable for model training. This data pipeline consists of four main stages: (1) raw data aggregation, (2) data type standardization and entity recognition, (3) modality extraction, and (4) training token sequence assembly.

### 2.1. Raw Data Aggregation

First, data from various sources are gathered into an object store in their raw, original form. This includes text from ArXiv and Wikipedia (Together.ai, 2023), assay data from ChEMBL (Gaulton et al., 2012) and the Therapeutic Data Commons (Huang et al., 2021), quantum mechanics and

[*]Equal contribution   [1]Iambic Therapeutics, San Diego, California, USA. Correspondence to: Matthew Welborn <matt@iambic.ai>.

*Accepted at the 1st Machine Learning for Life and Material Sciences Workshop at ICML 2024.* Copyright 2024 by the author(s).

*Table 1.* Pre-training Corpus.

| Source | Token Count (B) |
|---|---|
| **Text** | |
| peS2o (Soldaini & Lo, 2023) | 67.91 |
| arXiv (Together.ai, 2023) | 28.12 |
| Wikipedia (Together.ai, 2023) | 5.48 |
| BioRxiv (marianna, 2023) | 0.02 |
| PubMed & USPTO (Gao et al., 2020) | 72.98 |
| ClinicalTrials (U.S. National Library of Medicine, 2014) | 5.67 |
| **Assay Data** | |
| Biogen ADME (Fang et al., 2023) | 0.04 |
| Kinase200 (Luukkonen et al., 2023) | 0.77 |
| ChEMBL (Gaulton et al., 2012) | 60.80 |
| Internal assays | 0.46 |
| GOSTAR (Excelra, 2023) | 86.63 |
| Therapeutic Data Commons (Huang et al., 2021) | 4.85 |
| PubChem (Seidl et al., 2023) | 84.46 |
| **QM Data** | |
| Misato QM (Siebenmorgen et al., 2023) | 0.07 |
| Orbnet (Christensen et al., 2021) | 8.80 |
| **Knowledge Graph** | |
| PrimeKG (Chandak et al., 2023) | 3.39 |
| **Genetic Data** | |
| DepMap (Broad-Institute, 2020) | 73.76 |
| NCBI Gene Database (NIH, 2004) | 0.06 |
| **Protein Data** | |
| Protein Data Bank (wwp, 2019) | 2.89 |
| Misato MD (Siebenmorgen et al., 2023) | 1.28 |
| UniRef100 (Suzek et al., 2015) | 147.10 |
| **Synthetic** | |
| Randomized SMILES | 0.34 |
| Synthetic SMILES | 0.19 |
| Total | 656.07 |

molecular dynamics data from Misato (Siebenmorgen et al., 2023), genetic data from DepMap (Broad-Institute, 2020), protein structures from the Protein Data Bank (wwp, 2019), and protein sequences from Uniref100 (Suzek et al., 2015). In addition, synthetic datasets are generated corresponding to different protomer and tautomer states of molecules from ChEMBL as well as randomized SMILES strings using RDKit (Landrum, 2023). Table 1 provides the complete list of data sources included in the pre-training corpus.

## 2.2. Data Type Standardization and Entity Recognition

Second, each source of data is grouped by data type. For each data type, all data of that type are converted to a standard file format (e.g. protein structure data are converted to PDB format). Our data types include: assay descriptions, assay values, text, protein structures, molecular structures, molecule graphs, SMILES strings (Weininger, 1988a), FASTA sequences, knowledge graphs, JSON data, quantum mechanical data, cancer cell line data, gene dependency data, gene data, and molecular substance data.

During the same step, the data are tagged by the entities contained within them. Our entity types include: molecules, properties, assays, proteins, PDB codes, genes, diseases, symptoms, taxonomy, tissues, cell lines, chemical reactions, molecular functions, biological processes, cellular components, and chemical synthesis batches. For example, a protein-ligand structure datum containing the structure of

a CDK4–Cyclin-D3–abemaciclib complex is tagged with the protein entities CDK4 and Cyclin D3, the molecule entity abemaciclib, the PDB code 7SJ3, and the biological process mitosis. Entities are assigned using either metadata or named entity recognition techniques (Yoon et al., 2022). Additionally, numerical values corresponding to dimensioned quantities are transformed and converted to standard SI units.

## 2.3. Modality Extraction

Third, data are combined to form modality data. Modality data are 1D strings of characters constructed in a specific way for each modality in our multi-modal model. These modalities closely follow the data types above but are distinct in some ways. The modalities include: text, molecule graphs, molecule structures, atomic 3D coordinates, protein structures in PDB format, protein structures in 3Di encoding (van Kempen et al., 2022), protein ligand complexes in PDB format, protein sequences, nucleotide sequences, walks on knowledge graphs, knowledge graph neighborhoods, tabular data, and raw data files.

## 2.4. Training Token Sequence Assembly

Finally, modality data are converted to token sequences using a Byte-Pair Encoding (BPE) tokenizer (Sennrich et al., 2016) (Section 3.1). These token sequences are grouped into training samples on the basis of their shared entities (Section 2.2). Within a training sample, these token sequences are concatenated together, each preceded by a delimiter token indicating its modality. This scheme of grouping the token sequences is akin to RA-CM3 (Yasunaga et al., 2022). The set of all assembled token sequences are grouped into training data shards at random.

To prevent leakage of benchmark test data into pre-training, an entity holdout system is used. First, all of the entities corresponding to benchmark test data are gathered. Then, when assembling the pre-training token sequences, if a datum is tagged with an entity that is in the benchmark set of entities, it is excluded from the pre-training data. This system ensures that data from other sources closely related to the benchmark test data also do not leak into the pre-training corpus.

## 3. Model

### 3.1. Tokenizer

We start from the LLaMA-2 tokenizer which employs Byte Pair Encoding (BPE) (Sennrich et al., 2016) implemented in Sentence-Piece (Kudo & Richardson, 2018). We then add uni-gram tokens for numbers similar to FP15 format from Charton (2021), SMILES (Weininger, 1988b) tokens from

SmilesPE (Li & Fourches, 2021), and protein structural tokens from FoldSeek (van Kempen et al., 2022). See Supporting Information for an example token sequence which incorporates the special tokens described above.

## 3.2. Architecture

Our transformer model is based on the LLaMA-2 architecture (Touvron et al., 2023a;b). We make no changes and therefore use SwishGLU (Shazeer, 2020), Rotary Positional Encoding (RoPE) (Su et al., 2024), and Multi Head Attention (Vaswani et al., 2017). The model has a context length of 4096 tokens.

## 3.3. Pre-training

Our model implementation uses FLASH ATTENTION 2 (Dao et al., 2022; Dao, 2023), FUSED SWIGLU from xFormers (Lefaudeux et al., 2022), FULLY SHARDED DATA PARALLEL (FSDP) (Zhao et al., 2023) and automatic mixed precision training (Micikevicius et al., 2017) with `bfloat16`. Our production training runs use Docker images from the NVIDIA GPU Catalog (NGC), specifically the `23.09` tag of the PYTORCH (Paszke et al., 2017) Docker image.

Our model has approximately 1 billion parameters (Zhang et al., 2024) including token embedding and language modelling head parameters. We use MOSAICML STREAMING (Mosaic-ML-Team, 2022) for efficient sharded dataloading. We use 256 A100-40G NVIDIA GPUs to train the production model, totaling approximately 29,000 A100 hours of training time. The total training time includes time spent on validation steps during training, checkpoint saving and restarting from checkpoints between jobs. During the production training run we observe sustained training Model FLOPS utilization (MFU) of 0.61 using the PaLM estimation formula (Chowdhery et al., 2023). We use a global batch size of 4M tokens and AdamW optimizer (Loshchilov & Hutter, 2017) with $\beta_1 = 0.9$ and $\beta_2 = 0.95$, gradient clipping of 1.0 and weight decay of 0.1. We use cosine annealing schedule (Loshchilov & Hutter, 2017) with a maximum learning rate of $3.0 * 10^{-4}$ after an initial linear warm up of approximately 100B tokens.

We pre-train from scratch on the data curated in Section 2. Our pre-training objective is a mix of RA-CM3 (Yasunaga et al., 2022; Aghajanyan et al., 2022) and next token prediction (Radford et al., 2018; 2019; Brown et al., 2020). In total, we train the model using 2.25T tokens. We use the last checkpoint as the base model for benchmarking and downstream fine-tuning.

## 3.4. Fine-tuning

To fine-tune the model for assay prediction tasks, we use mean squared error (MSE) loss with minor changes to the

model architecture. The language modelling head is replaced with a regression head that has a fully connected layer with a single output (the numerical assay value prediction). The embedding of the last token in the sequence is used as the prompt embedding, which is fed to the regression head to generate a prediction.

Low-rank adaptation (LoRA) (Hu et al., 2021) is used for parameter-efficient fine-tuning of the base model using the PEFT library (Mangrulkar et al., 2022) with a rank of 16 and alpha of 16. We use the AdamW optimizer (Loshchilov & Hutter, 2017) with $\beta_1 = 0.9$ and $\beta_2 = 0.999$, a maximum learning rate of $1.0 * 10^{-4}$, a batch size of 8, and a cosine annealing learning rate scheduler with linear warm-up for the first 5% of total training steps.

Each benchmark test set is globally held out from training (both pre-training and fine-tuning). The best model is selected during fine-tuning which minimizes the MSE loss on the validation set (random 15% split of the training data).

## 3.5. Benchmarking

We benchmark our models on our own internal assays as well as 2 public benchmarks: Biogen ADME (Fang et al., 2023) and Kinase200 (Luukkonen et al., 2023). Biogen ADME contains 3,521 commercially available drug-like compounds measured by Biogen across 6 in-vitro ADME assays. Kinase200 is a large, but sparse, curated dataset of 216,858 kinase inhibition measurements for 198 kinases and 82,982 molecules. To save compute, we choose to benchmark only on the dissimilarity-driven global balanced clustering (DGBC) split of the 9 CDKs from Kinase200. Our own internal benchmark consists of tasks across 4 drug discovery domains including 6 absorption and distribution endpoints, 6 protein inhibition assays corresponding to internal drug targets, 2 physical chemistry properties, and 4 metabolic clearance tasks. See Supporting Information for detailed descriptions of all benchmark tasks.

When training on public datasets, data contamination is a foremost concern (Sainz et al., 2023). The Biogen ADME benchmark addresses data contamination by generating fresh experimental data; at the time of its publication, its 3,521 molecules had no known public measurements for its 6 ADME assays. Our internal benchmark contains data for multiple real-world drug discovery programs. Due to the proprietary nature of the molecules in this benchmark, we can be confident that they are excluded from the pre-training data corpus, and that the resulting benchmarks are not contaminated.

A total of 6 multi-task multi-modal transformer models are fine-tuned using the procedure described in Section 3.4: 1 for Biogen ADME, 1 for Kinase200 CDKs, and 4 for the internal benchmark (1 per domain). To obtain an estimate of

the uncertainty, performance metrics are average over five models trained with different random seeds. The reported uncertainty is the standard error of the mean (SEM $= \frac{s}{\sqrt{5}}$) over the five training runs.

These models are benchmarked against the base multi-modal transformer model and two external baselines: XG-Boost (Chen & Guestrin, 2016), a common tree-based method, and Chemprop (Heid et al., 2024), a state-of-the-art graph neural network. A single-task XGBoost model is trained on each task with a Morgan fingerprint (Rogers & Hahn, 2010) of size 2048 and radius 3. For Chemprop, single-task and multi-task models are trained using the default hyperparameters. For training all external baselines, the dataset is split 80/20 into training and validation partitions, and all models are benchmarked on the same held-out test set for each task. Predictions for the multi-modal base model are obtained using autoregressive greedy decoding. The model is prompted with the assay description and molecule SMILES.

# 4. Results & Discussion

## 4.1. Scaling Laws

*Table 2.* Mean Absolute Error (MAE) and Pearson correlation coefficient (Pearson R) of two base models trained on different numbers of tokens for the Biogen ADME benchmark test set.

| Task | 870B training tokens | | 2.25T training tokens | |
| --- | --- | --- | --- | --- |
| | MAE | Pearson R | MAE | Pearson R |
| HLM | 0.41 | 0.60 | **0.38** | **0.66** |
| HPPB | 0.87 | 0.48 | **0.66** | **0.71** |
| MDR1-MDCK-ER | **0.46** | **0.56** | 0.47 | 0.55 |
| RLM | **0.48** | **0.67** | 0.52 | 0.62 |
| RPPB | 0.82 | 0.55 | **0.56** | **0.74** |
| SOLUBILITY | 0.98 | -0.05 | **0.43** | **0.55** |

Transformer models are known to exhibit scaling laws (Hoffmann et al., 2022; Kaplan et al., 2020) in downstream benchmark performance, where increased training compute results in improved performance. In Table 2 we report performance on the Biogen ADME benchmark set for two base model checkpoints during pre-training. Specifically, we benchmark model checkpoints trained on approximately 870B tokens and 2.25T tokens. The model trained for 2.25T tokens significantly outperforms the model trained for 870B tokens in 4/6 tasks. In the remaining two tasks, the 870B-token model slightly outperforms the 2.25T-token model. This results is consistent with previous findings: while pre-training loss as a function of training compute behaves predictably, downstream performance is only correlated to training compute (Hoffmann et al., 2022; Du et al., 2024).

## 4.2. Benchmarks

*Table 3.* Pearson correlation coefficient (Pearson R) of the multi-modal transformer models trained in this work (Base and LoRA-finetuned) vs Chemprop and XGBoost on the Biogen ADME, Internal, and Kinase200 benchmarks. The error for Chemprop and LoRA is the standard error of the mean over 5 different random seeds. See Supporting Information for detailed descriptions of the tasks and a corresponding table reporting mean absolute errors.

| Task | XGBoost Single-Task | Chemprop Multi-Task | Chemprop Single-Task | This work Base | This work LoRA |
| --- | --- | --- | --- | --- | --- |
| | | Biogen ADME | | | |
| HLM | 0.559 | 0.743 ± 0.005 | 0.748 ± 0.004 | 0.656 | **0.813 ± 0.006** |
| HPPB | 0.397 | 0.707 ± 0.015 | 0.712 ± 0.010 | 0.707 | **0.822 ± 0.003** |
| MDR1-MDCK-ER | 0.608 | 0.741 ± 0.014 | 0.713 ± 0.011 | 0.554 | **0.821 ± 0.004** |
| RLM | 0.560 | 0.767 ± 0.005 | 0.736 ± 0.005 | 0.617 | **0.818 ± 0.003** |
| RPPB | 0.423 | 0.671 ± 0.032 | 0.688 ± 0.024 | 0.742 | **0.842 ± 0.004** |
| SOLUBILITY | 0.403 | 0.580 ± 0.012 | 0.582 ± 0.012 | 0.547 | **0.680 ± 0.005** |
| | | Internal absorption and distribution | | | |
| FBS-PB | 0.646 | **0.800 ± 0.006** | 0.751 ± 0.014 | 0.295 | 0.628 ± 0.028 |
| HLM-PB | 0.537 | **0.603 ± 0.019** | 0.418 ± 0.017 | 0.101 | 0.561 ± 0.016 |
| HPPB | 0.288 | 0.375 ± 0.051 | 0.236 ± 0.028 | 0.143 | **0.531 ± 0.044** |
| MDCK-MDR1-ER | 0.258 | 0.046 ± 0.035 | 0.207 ± 0.019 | -0.148 | **0.492 ± 0.036** |
| MPPB | 0.696 | 0.829 ± 0.011 | 0.677 ± 0.047 | 0.243 | **0.855 ± 0.004** |
| PAMPA PAPP | -0.057 | 0.115 ± 0.011 | 0.047 ± 0.084 | -0.050 | **0.617 ± 0.020** |
| | | Internal inhibition assays (IC50) | | | |
| PROTEIN 1 | **0.654** | **0.641 ± 0.022** | 0.584 ± 0.034 | 0.408 | **0.673 ± 0.016** |
| PROTEIN 2 | 0.491 | 0.641 ± 0.026 | **0.703 ± 0.020** | 0.413 | 0.670 ± 0.008 |
| PROTEIN 3 | 0.358 | 0.306 ± 0.023 | 0.342 ± 0.031 | 0.380 | **0.435 ± 0.015** |
| PROTEIN 4 | 0.607 | 0.626 ± 0.031 | 0.490 ± 0.075 | 0.374 | **0.677 ± 0.014** |
| PROTEIN 5 | 0.298 | 0.159 ± 0.021 | 0.235 ± 0.044 | 0.206 | **0.353 ± 0.016** |
| PROTEIN 6 | 0.483 | 0.550 ± 0.023 | 0.576 ± 0.021 | 0.516 | **0.628 ± 0.012** |
| | | Internal physical chemistry | | | |
| SOLUBILITY | 0.313 | 0.624 ± 0.008 | **0.629 ± 0.010** | 0.382 | **0.645 ± 0.009** |
| LOGD | 0.490 | **0.803 ± 0.009** | 0.737 ± 0.015 | 0.573 | **0.796 ± 0.018** |
| | | Internal metabolic clearance | | | |
| GSH | -0.410 | 0.029 ± 0.018 | -0.064 ± 0.027 | 0.175 | **0.307 ± 0.036** |
| HUMAN HEP | **0.365** | 0.145 ± 0.032 | **0.356 ± 0.060** | -0.246 | **0.458 ± 0.097** |
| MOUSE HEP | 0.073 | 0.128 ± 0.008 | 0.043 ± 0.012 | 0.107 | **0.255 ± 0.038** |
| RAT HEP | -0.214 | **0.219 ± 0.024** | 0.130 ± 0.016 | -0.148 | **0.269 ± 0.061** |
| | | Kinase200 | | | |
| *Median of 9 CDK tasks* | 0.240 | 0.412 | 0.383 | 0.275 | **0.565** |

The performance of the multi-modal transformer model (base and fine-tuned) pre-trained for 2.25T tokens is reported for all benchmarks in Table 3. Comparisons to XG-Boost and Chemprop are also presented. The base model performs poorly relative to XGBoost and Chemprop across most tasks. However, fine-tuning the base model dramatically improves its performance. The LoRA fine-tuned multi-modal transformer has the highest Pearson R on all Biogen ADME tasks and 10/18 internal assays. The fine-tuned model only underperforms Chemprop by a significant margin on 2 tasks: fetal bovine serum protein binding (FBS-PB) with a Pearson R of 0.628 vs 0.800 and human liver microsome protein binding (HLM-PB) with a Pearson R of 0.561 vs 0.603. It performs similarly to Chemprop on the internal physical chemistry tasks, and outperforms Chemprop and XGBoost on most internal absorption and distribution, protein inhibition, and clearance assays. The fine-tuned multi-modal model is also the most accurate model in predicting CDK inhibition for the Kinase200 benchmark. It has the highest median Pearson R and the lowest median mean absolute error across the 9 Kinase200 CDKs (see Supporting Information for a corresponding table reporting MAE values), demonstrating its capability to generalize over the deliberate chemical dissimilarity between the training and

test splits in Kinase200.

### 4.3. Comparison To Generalist LLMs

*Table 4.* Mean Absolute Error (MAE) and Pearson correlation co-efficient (Pearson R) of the LoRA fine-tuned multi-modal transformer model trained in this work vs a LoRA fine-tuned LLaMA-2 7B model on the Biogen ADME benchmark test set. The pre-trained model in this work outperforms on all tasks.

| Task | This work 1B LoRA | | LLaMA-2 7B LoRA | |
|---|---|---|---|---|
| | MAE | Pearson R | MAE | Pearson R |
| HLM | **0.279 ± 0.003** | **0.813 ± 0.006** | 0.333 | 0.751 |
| HPPB | **0.518 ± 0.008** | **0.822 ± 0.003** | 0.613 | 0.750 |
| MDR1-MDCK-ER | **0.317 ± 0.002** | **0.821 ± 0.004** | 0.381 | 0.771 |
| RLM | **0.319 ± 0.003** | **0.818 ± 0.003** | 0.374 | 0.754 |
| RPPB | **0.434 ± 0.007** | **0.842 ± 0.004** | 0.503 | 0.779 |
| SOLUBILITY | **0.330 ± 0.003** | **0.680 ± 0.005** | 0.342 | 0.648 |

We also test the effect of domain-specific pre-training by comparing to the generalist language model GPT-3.5 (GPT) and LLaMA-2 7B (Touvron et al., 2023b). We use OpenAI's fine-tuning API to fine-tune GPT-3.5-TURBO-0125 on the Biogen ADME dataset using prompts formatted as in Section 2. In comparison to both base and fine-tuned versions of our model, we find that the performance of fine-tuned GPT-3.5 is poor. Specifically, at a sampling temperature of 0, GPT-3.5-TURBO-0125 predicts a single value for each task, regardless of the input molecule. See the Supporting Information for more details.

We fine-tune LLaMA-2 7B on Biogen ADME using the same regression fine-tuning procedure and hyperparameters described in Section 3.4. Due to the increased training cost of fine-tuning a 7B parameter model, we only fine-tune a single multi-task LLaMA-2 model. Table 4 shows that, despite having fewer parameters, our 1B parameter pre-trained multi-modal model outperforms fine-tuned LLaMA-2 7B on all tasks. This highlights the significance of our domain-specific multi-modal pre-training in achieving state-of-the-art results on assay benchmarks.

### 4.4. Multi-Task Learning Curve

To investigate if the multi-modal multi-task transformer is an effective multi-task learner, we compute learning curves using different amounts of HLM and RLM training data for fine-tuning from the Biogen ADME benchmark. Specifically, we evaluate the Pearson correlation coefficient on the HLM test set when varying the amount of HLM training data from 50 to 200 samples and RLM training data from 0 to 200 samples. Figure 1 shows the performance on HLM increases as more RLM training data is added. The model achieves a Pearson R on HLM greater than 0.7 with 200 HLM data *or* only 50 HLM data and 200 RLM data. This demonstrates that the multi-modal multi-task transformer effectively leverages training data from multiple tasks (HLM

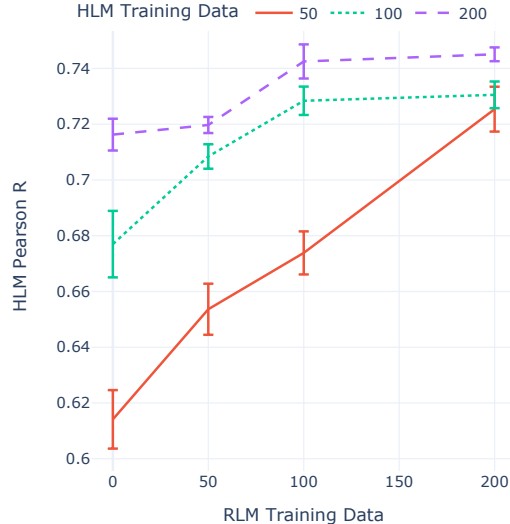

*Figure 1.* Biogen ADME HLM/RLM Multi-task Learning Curve. Adding RLM training data improves the model's performance on HLM.

and RLM) to improve performance in the low-data regime. This is critical in drug discovery programs where there may be an abundance of some data and shortage of more critical data that is expensive or difficult to obtain.

## 5. Conclusions

In this work, we have reported a multi-modal and multi-task transformer model trained on a large variety of biomedical data sources and data modalities. When benchmarked on assay prediction, this model outperforms representative standard techniques and generalist LLMs. We have also demonstrated how the model leverages multi-task data on metabolic stability in rat liver microsomes to improve predictive performance on human liver microsomes.

Assay prediction benchmarks are legible, common, and there are many ML techniques to address them. However, the multi-modal transformer reported herein generalizes naturally to other modalities and tasks. Future work could include predictions involving multiple chemical entities, such as reaction yields, and could further extend to include structured data, as in retrosynthesis prediction. The model, being generative, also lends itself naturally to generative tasks, including inverse design of molecules with desired properties.

Finally, it is well established that transformer models improve as the number of training data and model parameters

are increased. Our model is relatively small, and there is headroom to explore significantly larger models. In addition, the universe of public biomedical data is vast, and could readily provide orders of magnitude more training tokens.

## 6. Acknowledgments

This work used computational resources provided by the National Energy Research Scientific Computing Center (NERSC), under Contract No. ERCAP0029432.

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
