# Supporting Information: Multi-Modal and Multi-Task Transformer for Small Molecule Drug Discovery

Sai Krishna Sirumalla [* 1]   David S. Farina Jr [* 1]   Zhuoran Qiao [1]   Daniele A. Di Cesare [1]   Felipe Costas Farias [1]
Michael B. O'Connor [1]   Peter J. Bygrave [1]   Feizhi Ding [1]   Thomas Dresselhaus [1]   Marcelo G. P. de Lacerda [1]
Jason M. Swails [1]   Daniel Miles [1]   Matthew Welborn [1]   Frederick R. Manby [1]   Thomas F. Miller III [1]

[*]Equal contribution    [1]Iambic Therapeutics, San Diego, California, USA. Correspondence to: Matthew Welborn <matt@iambic.ai>.

*Accepted at the 1st Machine Learning for Life and Material Sciences Workshop at ICML 2024*. Copyright 2024 by the author(s).

# 1. Biogen ADME gpt-3.5-turbo-0125 results

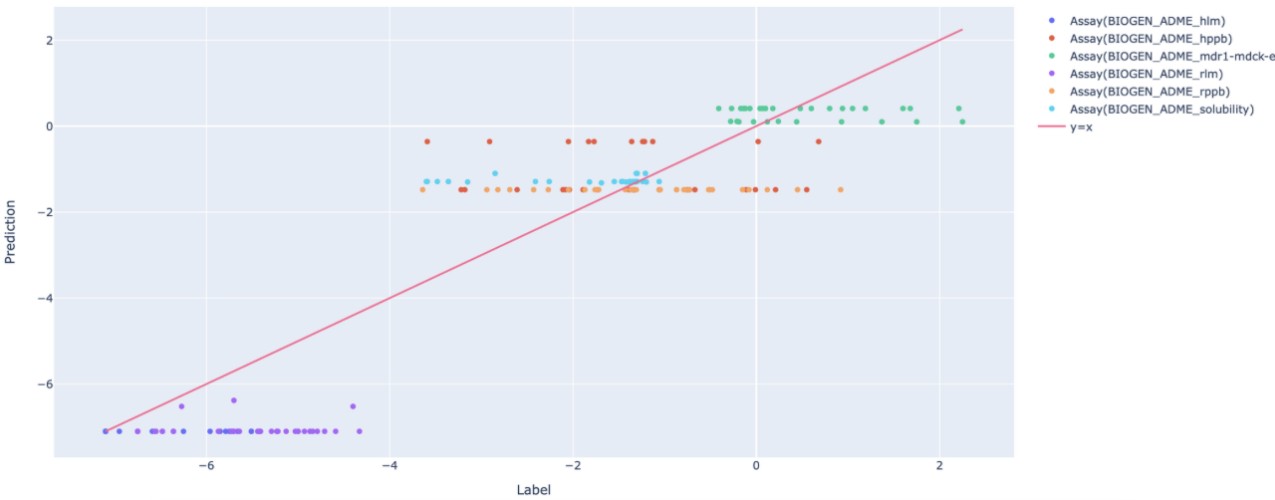

*Figure 1.* Scatter plot of predictions vs labels for a gpt-3.5-turbo-0125 model fine-tuned on the six Biogen ADME assays. Sampling is performed at zero temperature. Other temperatures were tested and were not found to improve accuracy.

# 2. Benchmark Tasks

The tables below describe the benchmark tasks used in this study. "Transform" describes the way in which data were transformed prior to training and testing. "log" refers to a natural logarithm, and "logit" refers to a logistic transform. "Num Train" and "Num Test" are the number of training and test data respectively. Each test datum contains one molecule identified by a SMILES string and an associated numerical label. All tasks are regression tasks.

"PROTEIN 1-6" refers to undisclosed protein targets of internal drug discovery programs.

*Table 1.*                                                                                                                          The
Internal Benchmark Tasks.

| Task | Description | Units | Transform | Num Train | Num Test |
|------|-------------|-------|-----------|-----------|----------|
| PROTEIN 1 | IC50 corresponding to the inhibition of protein target 1 by a ligand. | molar | log | 203 | 138 |
| PROTEIN 2 | IC50 corresponding to the inhibition of protein target 2 by a ligand. | molar | log | 203 | 142 |
| PROTEIN 3 | IC50 corresponding to the inhibition of protein target 3 by a ligand. | molar | log | 200 | 138 |
| PROTEIN 4 | IC50 corresponding to the inhibition of protein target 4 by a ligand. | molar | log | 166 | 108 |
| PROTEIN 5 | IC50 corresponding to the inhibition of protein target 5 by a ligand. | molar | log | 197 | 138 |
| PROTEIN 6 | IC50 corresponding to the inhibition of protein target 6 by a ligand. | molar | log | 109 | 103 |
| FBS-PB | Percent of compound unbound to fetal bovine serum | dimensionless | logit | 41 | 89 |
| GSH | Stability of a compound in presence of 5 mM glutathione. | second | log | 14 | 47 |
| HLM-PB | Percent of a compound unbound to human liver microsomes. | dimensionless | logit | 53 | 70 |
| HPPB | Percent of a compound unbound to human plasma. | dimensionless | logit | 27 | 47 |
| HUMAN HEP | Metabolic stability of a compound in human hepatocytes. | liter / cell / second | log | 54 | 93 |
| SOLUBILITY | Kinetic Solubility of a compound in a pH 7.4 PBS buffer. | molar | log | 337 | 295 |
| LOGD | Lipophilicity of a compound at a pH of 7.4 using the shake flask method. | dimensionless | log | 276 | 140 |
| MDCK-MDR1-ER | MDCK-MDR1 efflux ratio of a compound. | dimensionless | log | 52 | 133 |
| MOUSE HEP | Metabolic stability of a compound in mouse hepatocytes. | liter / cell / second | log | 70 | 151 |
| MPPB | Percent of a compound unbound to mouse plasma. | dimensionless | logit | 36 | 77 |
| PAMPA PAPP | Permeability of a compound through PAMPA membrane walls. | meter / second | log | 45 | 34 |
| RAT HEP | Metabolic stability of a compound in rat hepatocytes. | liter / cell / second | log | 47 | 83 |

*Table 2.*                                                               The
Biogen ADME Benchmark Tasks.

| Task | Description | Units | Transform | Num Train | Num Test |
|------|-------------|-------|-----------|-----------|----------|
| HLM | Metabolic stability of a compound in human liver microsomes | liter / gram / second | log | 2827 | 260 |
| HPPB | Measurement of the percent unbound of a compound to human plasma. | dimensionless | logit | 144 | 50 |
| MDR1-MDCK-ER | Measurement of MDR1-MDCK Efflux ratio. | dimensionless | log | 2419 | 223 |
| RLM | Metabolic Stability in rat liver microsomes | liter / gram / second | log | 2798 | 256 |
| RPPB | Measurement of the percent unbound of a compound to rat plasma. | dimensionless | logit | 118 | 50 |
| SOLUBILITY | Kinetic Solubility of a compound in a pH 6.8 PBS buffer. | atomic mass unit / bohr$^3$ | log | 1980 | 193 |

*Table 3.*                                                               The
Kinase200 Benchmark Tasks.

| Task | Description | Units | Transform | Num Train | Num Test |
|------|-------------|-------|-----------|-----------|----------|
| CDK1 | IC50 corresponding to the inhibition of CDK1 by a ligand. | molar | log | 949 | 110 |
| CDK2 | IC50 corresponding to the inhibition of CDK2 by a ligand. | molar | log | 6410 | 627 |
| CDK4 | IC50 corresponding to the inhibition of CDK4 by a ligand. | molar | log | 606 | 77 |
| CDK5 | IC50 corresponding to the inhibition of CDK5 by a ligand. | molar | log | 954 | 103 |
| CDK6 | IC50 corresponding to the inhibition of CDK6 by a ligand. | molar | log | 275 | 39 |
| CDK7 | IC50 corresponding to the inhibition of CDK7 by a ligand. | molar | log | 479 | 56 |
| CDK8 | IC50 corresponding to the inhibition of CDK8 by a ligand. | molar | log | 569 | 80 |
| CDK9 | IC50 corresponding to the inhibition of CDK9 by a ligand. | molar | log | 849 | 82 |
| CDK19 | IC50 corresponding to the inhibition of CDK19 by a ligand. | molar | log | 208 | 21 |

*Table 4.* Pearson correlation coefficient of the multi-modal transformer models trained in this work (Base and LoRA-finetuned) vs Chemprop and XGBoost on CDKs from the Kinase200 Benchmark

| | XGBoost Single-Task | Chemprop Multi-Task | Chemprop Single-Task | This work Base | This work LoRA |
|------|---------------------|---------------------|----------------------|----------------|----------------|
| CDK1 | 0.236 | **0.362 ± 0.035** | 0.188 ± 0.044 | 0.275 | **0.330 ± 0.038** |
| CDK2 | **0.540** | 0.418 ± 0.025 | 0.383 ± 0.012 | 0.261 | 0.480 ± 0.011 |
| CDK4 | 0.369 | 0.113 ± 0.101 | -0.228 ± 0.073 | 0.474 | **0.565 ± 0.019** |
| CDK5 | 0.240 | **0.447 ± 0.018** | 0.416 ± 0.014 | 0.263 | **0.420 ± 0.014** |
| CDK6 | -0.240 | 0.491 ± 0.060 | **0.600 ± 0.062** | -0.144 | **0.628 ± 0.025** |
| CDK7 | 0.315 | 0.412 ± 0.065 | 0.256 ± 0.040 | 0.144 | **0.577 ± 0.033** |
| CDK8 | 0.287 | **0.666 ± 0.016** | 0.552 ± 0.043 | 0.499 | **0.665 ± 0.013** |
| CDK9 | -0.056 | 0.078 ± 0.032 | -0.000 ± 0.026 | 0.325 | **0.387 ± 0.029** |
| CDK19 | 0.162 | 0.310 ± 0.022 | 0.533 ± 0.031 | 0.512 | **0.595 ± 0.010** |
| *Median* | 0.240 | 0.412 | 0.383 | 0.275 | **0.565** |

*Table 5.* Mean absolute error (MAE) of the multi-modal transformer models trained in this work (Base and LoRA-finetuned) vs Chemprop and XGBoost on CDKs from the Kinase200 Benchmark

|        | XGBoost Single-Task | Chemprop Multi-Task | Chemprop Single-Task | This work Base | This work LoRA |
|--------|---------------------|---------------------|----------------------|----------------|----------------|
| CDK1   | 0.537               | 0.542 ± 0.015       | 0.568 ± 0.015        | 0.849          | **0.509 ± 0.008** |
| CDK2   | **0.764**           | 0.884 ± 0.025       | 0.866 ± 0.011        | 0.922          | 0.773 ± 0.004  |
| CDK4   | 0.972               | 1.084 ± 0.053       | 1.332 ± 0.067        | **0.869**      | **0.880 ± 0.029** |
| CDK5   | 0.655               | 0.617 ± 0.013       | 0.694 ± 0.017        | 0.819          | **0.549 ± 0.011** |
| CDK6   | 1.279               | 0.982 ± 0.099       | 0.916 ± 0.052        | 1.137          | **0.796 ± 0.036** |
| CDK7   | 0.629               | 0.669 ± 0.049       | 0.665 ± 0.012        | 0.880          | **0.522 ± 0.020** |
| CDK8   | 1.083               | **0.826 ± 0.020**   | 0.961 ± 0.038        | 0.978          | **0.848 ± 0.013** |
| CDK9   | 0.941               | 0.973 ± 0.020       | 0.976 ± 0.014        | 0.869          | **0.746 ± 0.013** |
| CDK19  | 1.083               | 1.132 ± 0.018       | 0.964 ± 0.022        | 0.943          | **0.838 ± 0.019** |
| *Median* | 0.941             | 0.884               | 0.916                | 0.880          | **0.773**      |

*Table 6.* Mean absolute error (MAE) of the multi-modal transformer models trained in this work (Base and LoRA-finetuned) vs Chemprop and XGBoost on the Biogen ADME, Internal, and Kinase200 Benchmarks. The error for Chemprop and LoRA is the standard error of the mean over 5 different random seeds. The LoRA fine-tuned multi-modal transformer has the lowest MAE on 6/6 Biogen ADME tasks and 8/18 Internal tasks. It has the lowest median MAE on the Kinase200 benchmark

| Task | XGBoost Single-Task | Chemprop Multi-Task | Chemprop Single-Task | This work Base | This work LoRA |
|---|---|---|---|---|---|
| | | Biogen ADME | | | |
| HLM | 0.419 | 0.333 ± 0.002 | 0.330 ± 0.003 | 0.376 | **0.279 ± 0.003** |
| HPPB | 0.818 | 0.625 ± 0.014 | 0.616 ± 0.009 | 0.664 | **0.518 ± 0.008** |
| MDR1-MDCK-ER | 0.456 | 0.384 ± 0.013 | 0.398 ± 0.009 | 0.467 | **0.317 ± 0.002** |
| RLM | 0.476 | 0.373 ± 0.005 | 0.394 ± 0.005 | 0.519 | **0.319 ± 0.003** |
| RPPB | 0.735 | 0.584 ± 0.036 | 0.672 ± 0.013 | 0.562 | **0.434 ± 0.007** |
| SOLUBILITY | 0.445 | 0.392 ± 0.002 | 0.397 ± 0.005 | 0.434 | **0.330 ± 0.003** |
| | | Internal absorption and distribution | | | |
| FBS-PB | 0.375 | **0.293 ± 0.020** | 0.363 ± 0.007 | 0.472 | 0.361 ± 0.012 |
| HLM-PB | 0.357 | **0.337 ± 0.009** | 0.463 ± 0.006 | 0.911 | **0.330 ± 0.021** |
| HPPB | 0.659 | 0.480 ± 0.016 | 0.473 ± 0.004 | 0.962 | **0.412 ± 0.014** |
| MDCK-MDR1-ER | 0.473 | 0.555 ± 0.008 | 0.593 ± 0.005 | 2.085 | **0.421 ± 0.013** |
| MPPB | 0.655 | **0.408 ± 0.021** | 0.659 ± 0.014 | 1.275 | **0.420 ± 0.015** |
| PAMPA PAPP | 0.586 | 0.592 ± 0.013 | 0.527 ± 0.001 | 1.122 | **0.457 ± 0.022** |
| | | Internal Inhibition assays (IC50) | | | |
| PROTEIN 1 | **0.490** | 0.536 ± 0.009 | 0.582 ± 0.027 | 0.870 | **0.499 ± 0.010** |
| PROTEIN 2 | 0.805 | 0.658 ± 0.030 | **0.582 ± 0.015** | 0.923 | **0.605 ± 0.012** |
| PROTEIN 3 | 0.602 | 0.629 ± 0.011 | 0.598 ± 0.018 | 0.684 | **0.566 ± 0.005** |
| PROTEIN 4 | 0.646 | 0.676 ± 0.029 | 0.731 ± 0.033 | 0.848 | **0.616 ± 0.024** |
| PROTEIN 5 | **0.576** | 0.667 ± 0.011 | 0.623 ± 0.020 | 0.709 | 0.583 ± 0.005 |
| PROTEIN 6 | 0.636 | **0.580 ± 0.016** | **0.579 ± 0.019** | 0.641 | **0.562 ± 0.020** |
| | | Internal physical chemistry | | | |
| SOLUBILITY | 0.953 | 0.618 ± 0.015 | 0.616 ± 0.007 | 0.895 | **0.559 ± 0.010** |
| LOGD | 0.722 | **0.519 ± 0.010** | 0.577 ± 0.012 | 0.708 | **0.527 ± 0.019** |
| | | Internal metabolic clearance | | | |
| GSH | 0.867 | **0.631 ± 0.003** | **0.633 ± 0.005** | 0.805 | **0.612 ± 0.020** |
| HUMAN HEP | **0.502** | 0.545 ± 0.004 | 0.515 ± 0.002 | 0.843 | **0.488 ± 0.019** |
| MOUSE HEP | 0.446 | **0.381 ± 0.006** | **0.376 ± 0.001** | 0.643 | **0.394 ± 0.018** |
| RAT HEP | 0.555 | 0.500 ± 0.006 | 0.509 ± 0.003 | 0.874 | **0.474 ± 0.015** |
| | | Kinase200 | | | |
| *Median of 9 CDK tasks* | 0.941 | 0.884 | 0.916 | 0.880 | **0.773** |

## 3. Sample token sequence

```
<protein_structure_3di> <3Di>V</3Di><3Di>D</3Di> <3Di>F</3Di><3Di>C</3Di>
    <3Di>L</3Di><3Di>L</3Di> <3Di>A</3Di><3Di>A</3Di> <3Di>L</3Di><3Di>L
    </3Di> <3Di>V</3Di><3Di>L</3Di> <3Di>L</3Di><3Di>P</3Di> <3Di>P</3Di
    ><3Di>P</3Di> <3Di>D</3Di><3Di>D</3Di> <3Di>G</3Di><3Di>N</3Di> <3Di>
    L</3Di><3Di>S</3Di> <3Di>V</3Di><3Di>L</3Di> <3Di>S</3Di><3Di>S</3Di>
    <3Di>S</3Di><3Di>L</3Di> <3Di>Q</3Di><3Di>V</3Di> <3Di>S</3Di><3Di>L
    </3Di> <3Di>C</3Di><3Di>R</3Di> <3Di>Q</3Di><3Di>A</3Di> <3Di>V</3Di
    ><3Di>D</3Di> <3Di>G</3Di><3Di>Q</3Di> <3Di>Q</3Di><3Di>R</3Di> <3Di>
    A</3Di><3Di>P</3Di> <3Di>D</3Di><3Di>S</3Di> <3Di>V</3Di><3Di>S</3Di>
    <3Di>S</3Di><3Di>N</3Di> <3Di>V</3Di><3Di>S</3Di> <3Di>V</3Di><3Di>C
    </3Di> <3Di>Q</3Di><3Di>A</3Di> <3Di>A</3Di><3Di>W</3Di> <3Di>D</3Di
    ><3Di>D</3Di> <3Di>D</3Di><3Di>C</3Di> <3Di>V</3Di><3Di>Q</3Di> <3Di>
    A</3Di><3Di>N</3Di> <3Di>C</3Di><3Di>S</3Di> <3Di>C</3Di><3Di>V</3Di>
    <3Di>G</3Di><3Di>N</3Di> <3Di>N</3Di><3Di>R</3Di> <3Di>D</3Di><3Di>P
    </3Di> <3Di>P</3Di><3Di>D</3Di> <3Di>D</3Di><3Di>T</3Di> <3Di>H</3Di
    ><3Di>D</3Di> <3Di>H</3Di><3Di>D</3Di> <3Di>D</3Di><3Di>V</3Di> <3Di>
    C</3Di><3Di>V</3Di> <3Di>R</3Di><3Di>R</3Di> <3Di>P</3Di><3Di>P</3Di>
    <3Di>D</3Di><3Di>D</3Di> <3Di>P</3Di><3Di>Q</3Di> <3Di>S</3Di><3Di>S
    </3Di> <3Di>V</3Di><3Di>L</3Di> <3Di>S</3Di><3Di>V</3Di> <3Di>L</3Di
    ><3Di>S</3Di> <3Di>R</3Di><3Di>R</3Di> <3Di>N</3Di><3Di>H</3Di> <3Di>
    S</3Di><3Di>V</3Di> <3Di>S</3Di><3Di>R</3Di> <3Di>D</3Di><3Di>H</3Di>
    <3Di>S</3Di><3Di>V</3Di> <3Di>V</3Di><3Di>S</3Di> <3Di>S</3Di><3Di>V
    </3Di> <3Di>G</3Di><3Di>P</3Di> <3Di>S</3Di><3Di>C</3Di> <3Di>P</3Di
    ><3Di>P</3Di> <3Di>H</3Di><3Di>D</3Di> <3Di>R</3Di><3Di>D</3Di> <3Di>
    G</3Di><3Di>D</3Di> <3Di>D</3Di><3Di>D</3Di> <3Di>D</3Di><3Di>D</3Di>
    </protein_structure_3di> <SEP><tabular> [<< description >>] This
    assay assesses how effectively a drug candidate inhibits or modulates
     the activity of a protein kinase. It involves incubating the
    purified protein kinase with the drug candidate, its substrate, and
    ATP in a controlled environment. The reaction is initiated, and
    changes in the phosphorylation status of the substrate by the kinase
    in the presence of the drug candidate are quantified. The data
    obtained can aid in understanding the compound's specificity and
    efficacy in inhibiting the target kinase, which is vital for drug
    development and disease treatment. [<< assay_property >>] [<<
    category >>] potency [<< kind >>] biochemical [<< measurement >>]
    inhibition [<< protein >>] Cyclin-dependent kinase 2 (P24941) [<<
    kind >>] Literature [<< source >>] Kinase200 [<< units >>] molar [<<
    transform >>] log [<< assay >>] KINASE200_p24941_wt [<< SMILES >>] N#
     Cc1cn n2c( NC c3ccncc3) cc(-c3ccccc3) nc12 [<< value >>] 10^0*-6.21
    </tabular>
```

*Figure 2.* An example token sequence that includes two modalities: a 3D structure of CDK2 and an assay value for inhibition of CDK2 by 5-Phenyl-7-(pyridin-4-ylmethylamino)pyrazolo[1,5-a]pyrimidine-3-carbonitrile.