# OpenReview forum: "Multi-Modal and Multi-Task Transformer for Small Molecule Drug Discovery"
_ICML.cc/2024/Workshop/ML4LMS — ML4LMS Poster_

### Official Review · Reviewer_S3QT · 2024-06-12
**The paper excels in detailed experimentation but requires clearer task descriptions, given the model's broad applicability across diverse biomedical data modalities.**

**Rating:** 8
**Confidence:** 3

**Review:**

pros
The paper provides a comprehensive approach to experimentation
The methods employed are described in detail

cons
Some tasks are not clearly explained

suggestions
clarify lines 191-195
In Table 2, it is noted that the model trained on 2.25T tokens significantly outperforms the model trained on 870B tokens in 4 out of 6 tasks. It would be beneficial to discuss the differences between these 4 tasks and the other 2. Is this performance discrepancy expected? What factors might contribute to this variation?

---

### Official Review · Reviewer_rGjK · 2024-06-12
**Strong accept**

**Rating:** 9
**Confidence:** 4

**Review:**

The paper introduces a 1B parameter model based on LLaMA-2, trained over 15 modalities and 2.25T tokens, where all data are represented as strings of tokens using a data pipeline. The results presented in the paper show that their model finetuned with LoRA outperforms baselines such as ChemProp and XGBoost on Biogen ADME, Kinase200 CDKs and internal datasets. The paper is well written, providing details of the training conditions and hyperparameters, thus giving sufficient information for reproducibility. It is an interesting work covering 15 data modalities, would be interested in trying out the model if the authors plan to release the source code on gitHub.

---

### Official Review · Reviewer_QDjg · 2024-06-12
**Multi-modal pretraining for small molecule property prediction**

**Rating:** 8
**Confidence:** 4

**Review:**

The authors introduce a 1B parameter transformer, trained on multiple data modalities relating to drug discovery, and demonstrate that a training regime consisting of grouping similar token sequences based on shared named-entities allows their architecture to pretrain on these diverse modalities with minimal architectural tweaks or special encoders.

Pros:
- the authors take steps to prevent data leakage using an entity-holdout system in their pretraining pipeline, enhancing the reliability of the benchmarking & evaluation results.
- model, tokeniser & key hyperparameters are clearly described
- comparison against simple (XGBoost ensemble) and strong baselines (Chemprop) as well as comparison of base & finetuned versions of their model, which enables comparison of the effectiveness of pretraining
- the use of proprietary datasets for benchmarking does mean these molecules are unlikely to be present in the pretraining corpus and thus are more likely to indicate true generalisation

Cons:
- the authors present a simple data scaling analysis but do not examine or motivate their specific choice of a 1B parameter model, missing the opportunity to describe
- while the comparison of the finetuned models against the base model demonstrates the efficacy of the fine-tune, the importance of the pretraining itself is not compared to a model trained from scratch on the fine-tuning dataset
- the paper lacks an investigation as to the relative utility of various modalities and I would have been greatly interested in an analysis of which modalities can be removed from the pretraining pipeline while causing the least drop in downstream performance


Quality: the authors report relevant metrics to compare model performance across a range of downstream evaluations. The emphasis on preventing data leakage and use of entirely held-out data enhances the overall quality of the work.

Clarity: the motivation, methodology and results are clearly presented. The model, training and hyperparameters are well described, and the authors clearly present in table format their performance relative to relevant baselines on benchmark tasks.

Originality: the authors train a 1B parameter multi-modal transformer, using a novel pipeline and training token sequence assembler that enables multi-modal, multi-task learning. The RA-CM3 (retrieval augmented multi-modal language modelling) objective is used in addition to next-token prediction.

Significance: this work demonstrates a method for integrating multiple data modalities into sequences of tokens that can be used to train a transformer model at the 1 biilion parameter scale. The authors demonstrate the effectiveness of this pretraining scheme to create base-models that are able to outperform strong baselines after fine-tuning.

---

### Official Review · Reviewer_xEXr · 2024-06-12
**Multi-modal, multi-task approach for learning meaningful representations of molecules and their interactions**

**Rating:** 10
**Confidence:** 4

**Review:**

The authors train a multi-modal transformer of 1B parameters on 2.25T tokens. The dataset sources include text, assays, quantum mechanics, knowledge graphs, genetic data, protein sequences and synthetically generated small molecules. The authors tag the respective data with entities, such as genes, diseases, proteins, PDB codes. FInally they combine source to form modality data and convert token sequences using a BPE tokenizer, separating them with a delimiter token indicating the modality. Their model is based on the LLaMA-2 architecture and has a context length of 4096 tokens.

Pros:
- Used in-house assays as well as 2 public heldout benchmarks to validate the model
- Benchmarked the model against a strong ML baseline and a sota deep learning approach
- For the majority of their tasks, the model trained for longer significantly outperforms that other one
- Their finetuned model overall shows strong improvements over all baselines, however the pretrained model iitself s not very competitive

Cons:
-

Quality: very strong

Clarity: The motivation, methodology and results of the paper are clearly presented.

Originality: The multi-modality of their method is a novel approach to teaching a model about related chemical/biological concepts.

Significance: This work is a significant advancement from using well-structured, small datasets leading to strong improvements on various benchmarks.

---

### Official Review · Reviewer_4CJY · 2024-06-13
**Good paper highlighting the power of multi-modal pre-training.**

**Rating:** 9
**Confidence:** 4

**Review:**

Overall, this is a really good paper where the authors train a Multi-Modal and Multi-Task Transformer. Pretraining is performed on a wide range of datasets and then the authors perform low rank adaption for finetuning on specific datasets. it was really interesting to see that LORA and finetuning is required to actually get SOTA performance out of the model and that the performance improved with increasing number of tokens. The paper is well written and was easy to follow along.

My two minor nits with the paper would be one that the authors had used a tougher split for benchmarking than random and that they had used a chemistry specific pre-trained LLM such as chemberta etc as an additional baseline. Random splits are known to inflate performance in molecular modeling and it would be interesting to see how much using non-chemistry data helps with downstream performance.

However, overall, this paper is a clear accept for me.